# Prevalence of Leptospira Infection in Rodents from Bangladesh

**DOI:** 10.3390/ijerph16122113

**Published:** 2019-06-14

**Authors:** Inge M. Krijger, Ahmed A. A. Ahmed, Marga G. A. Goris, Peter W. G. Groot Koerkamp, Bastiaan G. Meerburg

**Affiliations:** 1Wageningen Livestock Research, Wageningen University & Research, P.O. Box 338, 6700 AH Wageningen, The Netherlands; bmeerburg@kad.nl; 2Farm Technology Group, Wageningen University & Research, P.O. Box 16, 6700 AA Wageningen, The Netherlands; peter.grootkoerkamp@wur.nl; 3World Organisation for Animal Health (OIE) and National Collaborating Centre for Reference and Research on Leptospirosis (NRL), Amsterdam University Medical Centers, University of Amsterdam, Medical Microbiology, Meibergdreef 39, 1105 AZ Amsterdam, The Netherlands; a.ahmed@amc.uva.nl (A.A.A.A.); m.goris@amc.uva.nl (M.G.A.G.); 4Dutch Pest and Wildlife Expertise Centre (KAD), Nudepark 145, 6702 DZ Wageningen, The Netherlands

**Keywords:** leptospirosis, rodents, reservoir, food safety, zoonosis

## Abstract

Worldwide, *Leptospira* infection poses an increasing public health problem. In 2008, leptospirosis was recognised as a re-emerging zoonosis of global importance with South-East Asia being one of the most significant centres of the disease. Rodents are thought to be the most important host for a variety of *Leptospira* serovars. Because Bangladesh offers a suitable humid climate for the survival of these pathogenic bacteria, the presence of rodents could be a serious risk for human infection, especially in peri-urban areas or locations where food is stored. In order to gain more understanding of the multi-host epidemiology, a prevalence study was conducted in Comilla, Bangladesh to determine the presence of pathogenic *Leptospira* species in rodents. Real-time Polymerase Chain Reaction (qPCR) and sequencing showed that 13.1% (61/465) of the trapped rodents were infected with pathogenic *Leptospira*. Sequencing of the qPCR products identified the presence of three species: *Leptospira interrogans*, *Leptospira borgpetersenii*, and *Leptospira kirschneri*. Rodents of the genus, *Bandicota*, were significantly more likely to be positive than those of the genus, *Rattus* and *Mus*. Our results confirm the importance of rodents as hosts of pathogenic *Leptospira* and indicate that human exposure to pathogenic *Leptospira* may be considerable, also in places where food (rice) is stored for longer times. This study emphasizes the need to improve rodent management at such locations and to further quantify the public health impacts of this neglected emerging zoonosis in Bangladesh.

## 1. Introduction

Commensal rodents are known to cause substantial pre- and postharvest losses. It is estimated that rodents contribute to 5% to 10% of the losses to rice production in Asia [1]. Besides causing direct loss to stored food, rodents also cause indirect loss: Their gnawing makes stored produce more prone to insect or fungal attacks and they contaminate a large percentage of produce with their droppings, urine, and saliva, which could possibly harbour pathogens [2,3,4,5,6]. A review by Meerburg et al. [3] points out the links between food security and rodents as rodents are potential reservoir hosts for over 60 zoonotic pathogens [7,8,9]. Asia is predisposed for (infectious) disease emergence [10] and there are numerous infection pathways of rodents with viruses, bacteria, and protozoans in Asia [11,12,13,14,15]. However, there is a limited number of studies available on the prevalence of rodent-borne diseases in many regions of Asia. This raises the need to determine pathogen prevalence, especially at locations where rodents come in close contact with humans or their stored food.

We studied a specific rodent-borne zoonotic pathogen, *Leptospira*, which is known to cause high disease burdens in Asia. This emerging spirochaetal bacteria occurs around rice agro-ecosystems with serious impacts on human health [1]. South-East Asia is mentioned as the most significant centre of the disease [16]. Studies led by the World Health Organization (WHO) on the global burden of human leptospirosis estimated more than 1 million severe cases with over 60,000 deaths annually [17,18,19,20,21]. Leptospirosis alone affects rural communities in most countries of Asia negatively, an endemic area for leptospirosis [16,22,23,24,25,26]. For example, in the rural areas of Bangladesh, there are innumerable ponds and shallow waters which facilitate the survival and transmission of the *Leptospira* to both maintenance hosts as well as dead-end hosts, such as humans.

In Thailand, the cases of human leptospirosis markedly increased over the 1995 to 2000 period. In 2000, leptospirosis was associated with 320 deaths reported among rice farmers [27]. This line is also seen in Malaysia, where the number of reported cases has multiplied over 14 times between 2004 (248 cases reported) and 2012 (3604 cases reported) [28]. This is even likely to be an underestimation because of the lack of awareness of leptospirosis symptoms due to the wide variety of these [29,30,31]. Moreover, it is expected that the global disease burden will increase due to climatic change in combination with population growth, the expansion of urban areas, and floods [3,32,33,34,35,36].

*Leptospira* is classified into 22 species, encompassing over 300 serovars [37,38], and in Bangladesh, at least 12 serovars have been observed [39]. Leptospirosis is maintained through chronic infection in the renal tubules of reservoir hosts, which shed *Leptospira* in their urine. The majority of mammalian species are natural hosts of pathogenic leptospires. Especially, small mammals can transmit infection directly or via contaminated water and food to domestic (farm) animals and humans [40,41,42]. Almost every rodent species may carry and excrete leptospires [43]. Rodents are thought to be the most important reservoir host for a variety of serovars, and serovar prevalence varies between rodent species [40]. Rats serve mostly as reservoir of the serovars, Icterohaemorrhagiae and Copenhageni, whereas serovars of the Ballum serogroup can be found in house mice (*Mus musculus*) [40,42,44,45]. *Leptospira* serovars usually do not cause disease in reservoir hosts, but do cause disease in the dead-end host, which in this case is the human [46]. Humans can acquire infection by contact with infected animals, animal tissue, animal excretions, or by contact of abrasions, cuts in the skin, or conjunctiva with contaminated water [47]. Leptospirosis causes feverish illness in humans, and when severe it can result in Weil’s disease [43]. Weil’s disease is characterized by jaundice, acute renal failure, and bleeding [48], and is often mistaken for several diagnoses [40,49,50,51].

An effective strategy to minimize infection is to limit contact between humans and commensal reservoir hosts. In Asian food production systems and storage, however, the risk of contact is almost unavoidable. Thus, it is essential to monitor infection prevalence of the reservoir, in order to target control of the reservoir host at times and places where the risk is highest [52]. Therefore, we aimed to gain insight in the prevalence of leptospirosis occurring in and potentially transmitted by rodents in Bangladesh.

Granaries offer a rich feed source for rodents, as well as suitable circumstances for the survival of leptospires. As several *Leptospira* reservoir hosts (e.g., rodents) live in the same locations as people work and produce food, granaries harbour a potential epidemiological niche for pathogen transmission to humans [53]. Information on leptospirosis and its consequences is extremely limited in many regions in Asia, amongst which is Bangladesh [27]. To our knowledge, this study is the first one conducted in Bangladesh on the infection of rodents with *Leptospira* spp. This knowledge gap results in a lack of precautions taken when handling commensal rodents or when preparing and consuming potential contaminated food [27], which requires actual prevalence rates of *Leptospira* in rodents living in and around food storages. The objective of this study was to assess the presence and infection rate of pathogenic *Leptospira* in commensal rodent species in rural Bangladesh.

## 2. Materials and Methods

### 2.1. Study Area

From March 2015 until March 2016, the first part of this study was conducted in a rice mill in the South-East region of Bangladesh. Two rice milling stations and 8 villages were selected for further research in 2016 and 2017. The villages were visited for 6 months per year: June, July, August, and October, November, December. Both rice mills were visited for a period of nine months, from August 2016 until March 2017. The selected rice mills and villages are situated in the Chittagong division, Comilla district (all within a 10 km circle from 23°27′23″ N 91°10′20″ E). Rodent trapping was conducted in the rice storage areas of the milling stations, where the paddy rice from the fields is stored in jute bags. The owners of the rice mills participated in the project and agreed to the use of their property and buildings for this study. In the villages, rodent trapping was conducted in the rice storage area of households. No ethics approval was required because the rodents that were trapped are pest species and the Bangladesh government has no regulations on animal ethics concerning pest species. However, all local staff were trained to work with the animals according to the Netherlands code of scientific practice and the participating researchers completed the Laboratory Science Course as required by European Directive 2010/63/EU and the revised Dutch Act on Animal Experimentation. Although the animals used were not laboratory animals, the NCad opinion on ‘Alternative methods for killing laboratory animals’ was followed, as provided by the Netherlands National Committee for the protection of animals used for scientific purposes. Moreover, the guidelines of the American Society of Mammologists were followed during the study. The above procedure was also mentioned in the original project proposal and approved by the donor (Netherlands Organization for Scientific Research, NWO-WOTRO).

### 2.2. Rodent Trapping

During the research period, every 14 days, rodent trapping was conducted for 3 consecutive periods of 24 h. Rodent trapping was performed by the use of 10 kill traps (14 × 7 cm; Big snap-e; Kness, Albia, IA, USA) and 10 life traps (purchased at local Bangladesh markets, Figure 1). Traps were placed during evening time at locations where rodent damage was observed and tracks were seen. Traps were checked by research staff the next morning. Rodents trapped in life traps were immediately euthanized by cervical dislocation and after identification of the species, gender, and life stage (juvenile/mature), each animal was dissected. Of each rodent, one kidney was randomly taken and stored in 98% ethanol and shipped to the OIE and National Collaborating Centre for Reference and Research on Leptospirosis (NRL), Amsterdam, The Netherlands for PCR testing.

### 2.3. Laboratory Diagnostics

Processing of the kidney samples for testing on pathogenic *Leptospira* was conducted at the NRL. From each kidney, a small transversal sample (up to 25 mg) was taken and processed for DNA extraction using the QIAamp DNA Mini Kit (QIAGEN, Hilden, Germany). A slightly adjusted protocol was applied; the amount of ATL buffer, proteinase K^+^, AL buffer, and absolute ethanol were doubled to ensure complete tissue lysis. Hereafter, DNA extraction took place according to the manufacturer’s protocol. The extracted DNA samples were stored at −20 °C until tested by real-time quantitative polymerase chain reaction (RT-qPCR). DNA samples were 1:10 diluted using UltraPure DNase/RNase-free distilled water (Invitrogen, UK) and tested in triplicate using SYBR Green real-time qPCR, targeting the *secY* gene [54] The reactions were set up to a final volume of 25 μL containing 10 μL of DNA sample; 1 μL of both forward and reverse primers, SecYIVF and SecYIV, at a final concentration of 400 nM each; 0.5 μL UltraPure DNase/RNase-free distilled water (Invitrogen, UK); and 12.50 μL of SYBR Green Supermix (Bio-Rad, UK) of 2 x stock reagent containing 100 mM KCl, 40 mM Tris-HCI, pH 8.4, 0.4 mM of each dNTP, 50 units/mL iTaq DNA polymerase, 6 mM MgCl_2_, 20 nM fluoresein, and stabilizers. As negative control, 10 μL of sterile UltraPure water was used. The reaction was performed and the result was analysed on a CFX96 real-time PCR detection system (Bio-Rad). The amplification protocol consisted of a first cycle of 5 min of activation at 95 °C, followed by 40 cycles of amplification (95 °C/5 s; 54 °C/5 s; 72 °C/15 s). The programme finished with 95 °C/min and cooling at 20 °C/1 min, and the amplified product was melted (70–94 °C) with plate readings set at 0.5 °C. A melting curve analysis was conducted to check the amplicon specificity. Samples were classified as positive when Ct values were ≤35 cycles with a Tm value between 78.5 and 84.5 °C. Samples were tested in triplicate and classified as positive when ≥2 runs resulted as positive. Samples that showed one amplification curve were retested in triplicate and classified as positive if the repeat run resulted in ≥1 positive reaction, and if this reaction showed an amplification melting curve that conformed to the set values. All products’ real-time PCR analysis were sequenced (Macrogen, Seoul, Korea), regardless of the outcome of the PCR and blasted to double-check PCR-results. Blast results were accounted for as decisive.

### 2.4. Statistical Analysis

Descriptive statistics were used, and for comparisons between rodent species, Fisher’s exact test was conducted. Results were considered statistically significant with a *p*-value of *p* < 0.05. Statistical analyses were performed by using SPSS, version 23 (IBM SPSS Statistics Inc, College Station, TX, USA). 

## 3. Results

### 3.1. Rodent Trapping

In total, 465 rodents were trapped in the rice mills and villages. Most trapped rodents were identified as *Bandicota bengalensis* (*n* = 140), *Rattus rattus* (*n* = 191), followed by *Mus musculus* (*n* = 97), *Rattus exulans* (*n* = 23), *Bandicota indica* (*n* = 9), and *Mus terricolor* (*n* = 5). In the dry season, more rodents were trapped (*n* = 292) compared to the rainy season (*n* = 173).

### 3.2. Laboratory

No anomalies were found in the animals during the dissections. Out of 465 rodents, 177 rodents showed a positive qPCR result. Sequencing of the samples from the qPCR run revealed 61 samples which showed sequence data of the partial *secY* gene upon alignment. Table 1 shows these real-time PCR and sequencing results (more detailed information on all 61 positive samples can be found in Appendix A).

In total, 13.1% (61/465; SD = 0.33) of the tested rodents were infected with pathogenic leptospires (Table 2). Of the five *Mus terricolor* animals that were trapped, none were positive for *Leptospira.* The other five rodent species tested did show positive samples, showing a significant difference between infection rates per species (*p* < 0.000). The highest infection rate was found in *B. indica* (77.8%, 95% CI = 0.43–1.1), followed by *R. exulans* (34.8%, 95% CI = 0.14–0.55), *B. bengalensis* (18.6%, 95% CI = 0.12–0.25), *R. rattus* (7.9%, 95% CI = 0.04–0.11), and *M. musculus* (5.5%, 95% CI = 0.01–0.09).

When looking at gender, in total, the kidney samples of 33 (14.2%) of 233 (Standard Error = 0.02) female rodents and 28 (12.1%) of 232 (SE = 0.02) male rodents were positive for pathogenic leptospires by sequencing and qPCR. Only female *R. rattus* were significantly more likely to be positive for *Leptospira* compared to male *R. rattus* (a *p*-value of 0.01, Pearson Chi-Square).

Significant differences were found when analysing for infection probability between species: *B. indica* showed significantly higher infection rates with *Leptospira* than all other species (*p* < 0.05, two tailed Pearson Chi-Square, Figure 2), and *B. bengalensis* and *R. exulans* both showed significant higher infection rates than *M. musculus* and *R. rattus*.

The obtained sequence data indicated three different types of pathogenic *Leptospira* strains present in the rodents: *Leptospira interrogans* (*n* = 15, GenBank accession no: CP020414.1), *Leptospira borgpetersenii* (*n* = 11, GenBank accession no: CP015814.2), and *Leptospira kirschneri* (GenBank accession no: LSSQ00000000.1). No significant link between the rodent species and encountered *Leptospira* strains was found (Table 3). 

When considering the two seasons (wet and dry), a significant difference was found for the effect of season (wet/dry) on the chance of infection with *Leptospira* (*p* = 0.019, two tailed Pearson Chi-Square, Table 4), with a higher chance of infection in the dry season.

## 4. Discussion

Pathogenic *L. interrogans*, *L. borgpetersenii*, and *L. kirschneri* species were identified in the rural Bangladesh rodent population. Almost 1 out of every 7 (61/465) rodents trapped had *Leptospira* bacteria in their kidneys and where thus potentially capable of shedding leptospires in and around food storage. In our study, no damage or anomalies were found during the dissections, which confirms the role of the animals as a natural reservoir. These results indicate that the risk of human exposure to pathogenic *Leptospira* is likely to be substantial for the workers of the Rice Milling Station of Comilla, and also for local people, since people can acquire leptospirosis via direct or indirect contact with the urine of an infected host, which in this case can also lead to a risk for the consumers of rice. However, although handling the rice may be dangerous, the risk of contracting leptospirosis via food consumption is limited if the rice is properly cooked.

The prevalence of leptospirosis in humans in Thailand, Malaysia, and India has been reported for some areas with infection rates between 15% and 35% [55,56,57,58,59,60,61,62,63,64]. Despite the occurrence of leptospirosis in South-East Asia, only a few studies have been performed on *Leptospira* prevalence in rodents, and to our knowledge, no studies from Bangladesh have been published before. Moreover, there are only scarce and frequently dated reports about the epidemiology of leptospirosis among citizens in Bangladesh. Research from 1994 showed a human seroprevalence of 38% (*n* = 89) in a rural district of Bangladesh close to rivers that regularly flood [39]. However, no link with risk factors (such as rodents) was made. In 2001, febrile patients (*n* = 1297) from two hospitals in Dhaka, Bangladesh were tested on leptospirosis and 63 patients (4.8%) were found to be positive [23]. More recent research from Bangladesh showed that over 13% of febrile people (*n* = 584) were serologically positive for *Leptospira* organisms [65]. One study on the prevalence of *Leptospira* in Bangladesh looked at dairy cows in Chittagong and showed that almost 50% of the samples were positive for *Leptospira* organisms [66], which poses a high infection risk to people working on cattle farms.

A study from Cambodia (2012) on *Leptospira* in rodents showed an overall infection of 11.1% (*n* = 642) [67] and a study from Malaysia found 11.0% (*n* = 357) [68], which correlates with our findings (13.1%), although we did not use a serological assay but a molecular assay. A serological study from Vietnam showed that 22% of trapped rodents host *L. interrogans* and all rodents were trapped in urban areas close to the South-China Sea and in Hanoi City, a city along the Red River [69]. Research from 2003 conducted on the Andaman Islands showed a seroprevalence of 7.1% in *R. rattus* (*n* = 85) [70], which again is in line with our findings (7.9%). Research on rodents from a suburban area in India showed 14.3% of the trapped *R. rattus* (*n* = 28), and 16.1% of the *B. bengalensis* (*n* = 58) were serological positive for *Leptospira* [71]. The findings specifically on *R. rattus* infections in India by Saravanan et al. were higher than the 7.9% found in the *R. rattus* (*n* = 191) from the current study in Bangladesh. This difference could be caused by the sensitivity and specificity of the used testing methods (serology vs. molecular diagnostic), or differences in the environment, to the ratio in which rodent species occurred and possibly also to the trapping easiness (or shyness) of each species. Meta-studies on *Leptospira* prevalence in rodents in South-East Asia (Table 5) showed that *B. bengalensis, B. indica, B. savilei,* and *R. exulans* were infected with *Leptospira*, whereas investigated species of the *Mus* genus appeared to be uninfected [60,61,72,73]. In contrast, we found a *Leptospira* prevalence of 5.5% in trapped *M. musculus*.

For *B. bengalensis* and *R. rattus*, the ratio of males and females trapped was almost evenly distributed (≈1:1). For *R. exulans* (although we trapped only 23 animals of this species), more males than females were trapped (18 out of 23). For the other three species from this study, *B. indica, M. musculus*, and *M. terricolor*, more female specimens were collected. Research in 2006 from Pakistan found 40% of the trapped *B. bengalensis* to be female (*n* = 167). This male:female ratio can, in comparison to our results, be explained due to the fact that rodents trapped in this study in Bangladesh were trapped at indoor locations only, whilst in the study of Rana et al. [74] they were trapped over multiple crop fields, which reflects the difference between male and female behaviour more. Males are more active in their explorations and depredation on food crops, whereas females may be more (trap) shy or are more concealed, or were retracted prior to breeding periods [74]. Moreover, in our study, there was no correlation between *Leptospira* prevalence and the gender of the animals for five of the six rodent species, which was also demonstrated in other earlier studies [75,76]. Some studies have shown that in Norway rats (*R. rattus*), both in South-East Asia and in the United States of America, male specimens have a higher infection rate than females [77,78,79]. However, in the current study, it was found that female *R. rattus* were significantly more prone to a *Leptospira* infection than males. Apparently, in our situation, the habitat’s use of rats creates a mechanism where both sexes were more or less equally exposed to infection. 

When looking at the results from the rodent population from our study sites, it showed *R. exulans* (*n* = 23) were present in lower numbers compared to both *B. bengalensis* (*n* = 140) and *R. rattus* (*n* = 191), while the population of *R. exulans* consisted mainly of males. This raises the question of whether *R. exulans* competes with one of the other rodent species present. It is not unusual that species compete with each other. An early report from Bombay and Calcutta (1966) showed that *B. bengalensis* increased in population size enormously and displaced the *R. rattus* in urban areas [80]. One of the underlying mechanisms was the high reproductive capacity of *B. bengalensis* [81,82,83], and the aggressive behaviour that dominant males exhibit [82,83]. Other research from India reported *B. bengalensis* to be the most aggressive field rodent [84] and that the females confine most of their time to hoarding food and are less active than males [83]. Furthermore, although bandicoots are generally nocturnal, the *B. bengalensis* is known to also become active during daytime when conditions are undisturbed [83]. This could be advantageous in comparison to other rodent species. Unfortunately, almost no scientific studies are published on the competition of *B. bengalensis* with other rodent species. We found only one report from India that claimed that female *R. rattus* are submissive to *B. bengalensis* [85]. Sridhara et al. [85] postulate that there is less aggression in closely related rodent species (e.g., within species from the *Rattus* genus) compared to the violent interaction of rodents more distantly related from each other (e.g., *Rattus* genus vs. *Bandicota* genus). Unfortunately, rodents of the *Rattus exulans* species were not researched. This finding might explain the higher number of *Bandicota* specimens and the higher *Leptospira* infection rate compared to *Rattus* specimens trapped in our study. *R. rattus* is amongst the most omnipresent rodents in the world, and has a strong potential to displace other (native) rodent species [86,87]. In Australian ecosystems, it was shown that the invasive *R. rattus* was dominant over the native *R. fuscipes* [88,89]. On Madagascar, *R. rattus* competes for resources with the native rodents and replaces native rodent species [90]. Because *B. bengalensis* and *R. rattus* are larger than *R. exulans* [91], we expect *B. bengalensis* and *R. rattus* to be superior to *R. exulans*. In New Caledonia, Perez et al. [92] most frequently trapped *R. Rattus* specimens (>60%, *n* = 140) and very rarely trapped *R. exulans* rodents (<5%, *n* = 11), which supports our hypothesis that *R. rattus* is dominant over *R. exulans*.

In Bangladesh, *Leptospira* prevalence (in %) in rodents was significantly higher in the dry season than in the rainy season. These findings are not in line with the findings from Malaysia and also from Cambodia, where rodents showed a lower infection rate in the dry (6.3%) than in the wet (26.7%) season [67,68]. In a study on febrile patients (*n* = 1297) from Dhaka, a peak in the occurrence of *Leptospira* in patients was found in October and November, shortly after the monsoon season in Bangladesh [23]. These findings confirm the relation between floods or excessive rainfall and Leptospirosis outbreaks [21,32,70]. Besides this seasonal influence on human infection, the risk of *Leptospira* infection also depends on the geographic location, as well as on other risk factors, such as the risk of flooding, contaminated surface waters, and proximity to rubbish dumps (attractive for rodents) [93,94,95]. Easterbrook et al. [77] stated that seasonal fluctuations in *Leptospira* infections in rodents do not occur due to the fact that once infected, the antibodies remain in the animal and the animal will test positive. This can explain our results and findings from other countries in South-East Asia, which show that rodent species living in households have a stable infection level, regardless of the geography and season [67,78].

Our study has some limitations. The extracted DNA from the kidneys was diluted before being added to the PCR mix in order to obtain better results by reducing potential inhibitors present in the samples. Inhibitors reduce the activity of the DNA polymerase enzyme and as a result a false negative result will be observed. As pathogenic *Leptospira* colonize the kidney of the rodent, using this tissue as a source for leptospires DNA is critical; however, very high concentrations of host DNA are present. The abundant DNA of the host in the PCR reaction could lead to false-positive results as SYBR green dye is able to bind to any double-stranded DNA present in a very high concentration, thus rodent DNA can also be bound. By sequencing all samples, it was possible to filter out all false positive qPCR results. Furthermore, the effect of potential residual inhibitors in the samples could possibly result in false negative qPCR results. Because of this, we consider the 13.1% prevalence to be a prudent estimate of the actual prevalence in the rodent population in the study areas in Comilla. However, the asset of this study is that molecular diagnostics were used instead, which gives an indication of the carriership of the animal. Serologic methods used by most of the other studies conducted on *Leptospira* in South-East Asia indicate that the animal did have contact with leptospires, but it remains unclear whether the animal is still a possible carrier and therefore a potential reservoir [27,60]. 

We found *L. interrogans* as well as *L. borgpetersenii* DNA in the samples from all five rodent species that tested positive (*B. bengalensis*, *B. indica*, *M. musculus*, *R. rattus*, *R. exulans*). Other studies confirm the relation between *R. exulans* and *L. interrogans* [78,92], and in Thailand, *R. exulans* was also found to be infected with both *L. interrogans* and *L. borgpetersenii* [60,96]. These findings are in line with the fact that *Leptospira* species, *borgpetersenii* and *interrogans*, contribute a great deal to human disease in Asia [49,61,78,79]. Also, in Europe, *L. borgpetersenii* and *L. interrogans* are the most observed *Leptospira* genomospecies present in rodents; however, in Europe, a third genomospecies is also commonly found in rodents: *L. kirschneri* [97,98]. It is interesting to note that we also found one rodent sample infected with *L. kirschneri* from a *R. exulans* sample. Our findings are the first to confirm the presence of *L. kirschneri* in the rodent species, *R. exulans*, in Bangladesh.

*Bandicota* is an Asian genus of rodents, consisting of three species: *B. bengalensis*, *B. indica*, and *B. savilei* [103,104,105,106,107]. The few reports on *Leptospira* prevalence in the *Bandicota* genus indicate that all three species are potential carriers of the same leptospira species (*L. interrogans, L. borgpetersenii, L. weilli, L. inadai*). Our study found that the probability of an infection with *Leptospira* was significantly higher for rodents of the *Bandicota* genus. Thus, *Bandicota* rats could be an important host in the epidemiology of leptospirosis in Comilla. Previous studies on urban rodents identify *R. rattus* to be the main reservoir host for human pathogenic *Leptospira* [94,108,109]. However, due to the limited information available, it is not possible to link the strain or serovar infection to a specific host species. This is unfortunate, as such information could give insight into a possible co-evolution of serovars with specific rodent species. Other studies from countries with a similar climate and cropping season as Bangladesh have used mostly serological and culturing methods, and unfortunately no specification is made into specific serogroups for each rodent species. Therefore, the only comparison possible is to see which strains are found and if this correlates with the findings in Bangladesh (Table 5).

Cosson et al. [78] postulate that *Leptospira* species show an ecological niche; they found *L. borgpetersenii* to be more abundant in rodents from dry habitats (non-floodable lands) than *L. interrogans,* which implies that the infection of rodents can be linked to ecology. The *B. indica* (*n* = 172) is more common in the field (74%) than in or around houses [110,111], and in Vietnam, the *B. bengalensis* was found only in grass habitats (1969). The fact that the *Bandicota* species from Bangladesh were found to be infected with *L. interrogans* (Table 5) is thus in line with expectations.

Our study provides new data on rodent species as carriers of pathogenic *Leptospira* in South-East Asia. From our results and the literature research (Table 5), we can state that *L. interrogans* and *L. borgpetersenii* are the most common species found in rodents in South-East Asia. However, to find out whether specific strains/serovars adapt to specific reservoir hosts in specific habitats, more in-depth research with different diagnostics needs to be conducted. Although our results confirmed the importance of *Bandicota* spp. and *Rattus* spp. as hosts of leptospires for human health and our findings indicate that human exposure to pathogenic *Leptospira* may be considerable in Comilla, the impact of leptospires on human health continues to be under recognised. In many Asian human populations, including populations in Bangladesh, the burden of undifferentiated feverish illness is substantial [26]. One way to minimise this problem of recognising the disease is to conduct broader diagnostic tests to determine the cause of these illnesses and to inform people on the preventive measures they can take to prevent leptospirosis. Our findings highlight the complex multi-host epidemiology of leptospirosis and the importance of considering the role of rodents, and other animal hosts in the maintenance and transmission of infection when evaluating human risks. One of the key actions to minimise the public health impacts of leptospirosis in Bangladesh is to improve rodent management. A key question is the percentage to which the rodent population should be reduced to and which species should be diminished to prevent infection. In any case, preventive measures should be taken for rodent control, such as preventing rodents from accessing domestic areas and food storage to prevent pathogen transmission to humans. Rodents could be denied access to food and drinking water by taking rodent-proofing measures to existing buildings or by constructing rodent-proof warehouses. Furthermore, rodents could be discouraged from visiting domestic areas by keeping the environments clean, removing potential nesting sites, and by installing adequate sanitation and waste disposal.

With more knowledge about the rodent species present, leptospirosis, and its consequences, local people can be informed about the need for better rodent management practices. This could lead to a reduction of the impact of rodent-borne zoonotic disease in Bangladesh.

## 5. Conclusions

The objective of this study was to assess the presence and infection rate of pathogenic *Leptospira* in commensal rodent species in rural Bangladesh. In order to do so, 465 rodents were collected from a total of six different species, in descending order of appearance: *Bandicota bengalensis*, *Rattus rattus*, *Mus musculus*, *Rattus exulans*, *Bandicota indica*, and *Mus terricolor*.

Pathogenic *Leptospira* was found in 13.1% of all rodents, and three *Leptospira* species were identified: *Leptospira interrogans, Leptospira borgpetersenii,* and *Leptospira kirschneri.* Significant differences were found for the infection probability between species: *B. indica* showed significantly higher infection rates with *Leptospira* than all other species, and *B. bengalensis* and *R. exulans* both showed significantly higher infection rates than *M. musculus* and *R. rattus*.

Rodents trapped carrying *Leptospira* bacteria in their kidneys (13.1%) are potentially capable of shedding leptospires in and around food storage. These findings indicate that the risk of human exposure to pathogenic *Leptospira* is likely to be substantial for local people, since people can acquire leptospirosis via direct or indirect contact with the urine of an infected host, which in this case can also lead to a risk for the consumers of rice. We can conclude that our findings highlight the complex multi-host epidemiology of leptospirosis and the importance of considering the role of rodents, and other animal hosts, in the maintenance and transmission of infection when evaluating human risk. One of the key actions to minimise the public health impacts of leptospirosis in Bangladesh is to improve rodent management.

## Figures and Tables

**Figure 1 ijerph-16-02113-f001:**
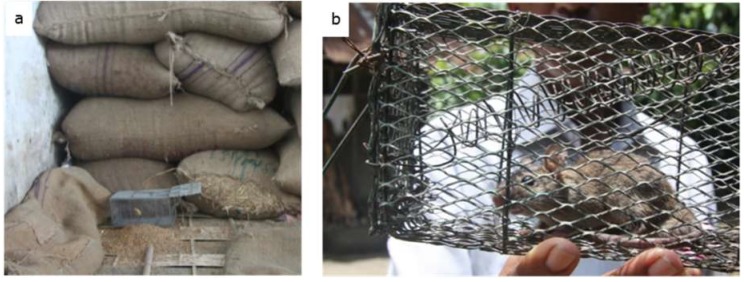
Example of (**a**) the placement of a life-trap in a rice milling station, and (**b**) a rodent trapped in a locally purchased life trap.

**Figure 2 ijerph-16-02113-f002:**
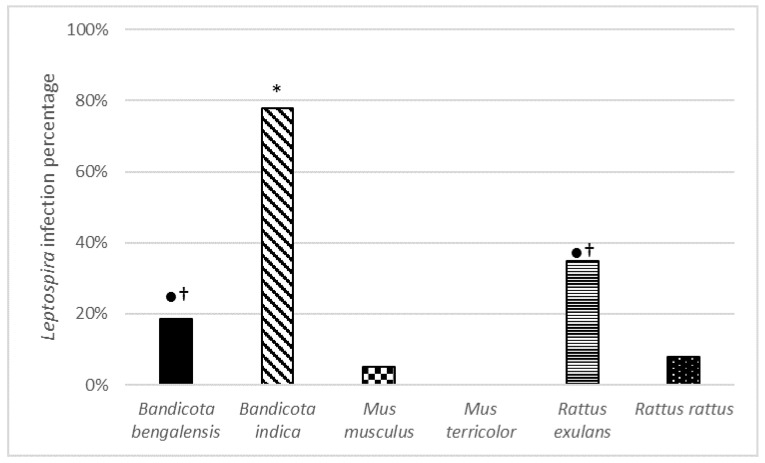
Infection percentage of six commensal rodent species from Bangladesh with *Leptospira*. * Significant difference from all (*p* < 0.05), † significant difference from *M. musculus* (*p* < 0.005), ● significant difference from *R. rattus* (*p* < 0.005).

**Table 1 ijerph-16-02113-t001:** Number of positive and negative tested kidney tissue samples from six commensal rodent species from Bangladesh for two types of tests to identify *Leptospira*, and the total number. Positive numbers are followed by their relative number (%).

Rodent Species	qPCR		DNA Sequencing		Total
	+	−	+	−	
*Bandicota bengalensis*	44 (31.4)	96	26 (18.6)	114	140
*Bandicota indica*	6 (66.7)	3	7 (77.8)	2	9
*Mus musculus*	53 (54.6)	44	5 (5.2)	92	97
*Mus terricolor*	0 (0.0)	5	0 (0.0)	5	5
*Rattus exulans*	19 (82.6)	4	8 (34.8)	15	23
*Rattus rattus*	55 (28.8)	136	15 (7.9)	176	191
Total	177 (38.1)	288	61 (13.1)	404	465

**Table 2 ijerph-16-02113-t002:** Prevalence of *Leptospira* infection (positive/total number and relative number in % in parentheses) determined by sequencing and qPCR among six different rodent species per gender and total.

Rodent Species	Prevalence		Total
	Female	Male	
*Bandicota bengalensis*	13/67 (19.4)	13/73 (17.8)	26/140 (18.6)
*Bandicota indica*	5/6 (83.3)	2/3 (66.7)	7/9 (77.8)
*Mus musculus*	2/58 (3.4)	3/39 (7.7)	5/97 (5.5)
*Mus terricolor*	0/4 (0.0)	0/1 (0.0)	0/5 (0.0)
*Rattus exulans*	1/5 (20.0)	7/18 (38.9)	8/23 (34.8)
*Rattus rattus*	12/93 (12.9)	3/98 (3.1)	15/191 (7.9)
Total	33/233 (14.2)	28/232 (12.1)	61/465 (13.1)

**Table 3 ijerph-16-02113-t003:** Number of rodents found positive for six rodent species and three *Leptospira* species using qPCR and sequencing.

Rodent Species (*n*)	Number of Rodents Positive		
	*L. borgpetersenii*	*L. interrogans*	*L. kirschneri*
*Bandicota bengalensis* (140)	10	16	0
*Bandicota indica* (9)	2	5	0
*Mus musculus* (97)	1	4	0
*Mus terricolor* (5)	0	0	0
*Rattus exulans* (23)	1	6	1
*Rattus rattus* (191)	5	10	0
Total	19	41	1

**Table 4 ijerph-16-02113-t004:** Prevalence of *Leptospira* infection (number infected/total and %) determined by sequencing and qPCR among six different rodent species and total divided into the dry and wet seasons.

Species	Dry Season	Wet Season
*Bandicota bengalensis*	19/77 (2.5)	7/63 (11.1)
*Bandicota indica*	7/9 (77.8)	0/0 (0)
*Mus musculus*	3/64 (4.7)	2/33 (6.1)
*Mus Terricolor*	0/4 (0)	0/1 (0)
*Rattus exulans*	7/12 (58.3)	1/11(9.1)
*Rattus rattus*	10/126 (7.9)	5/65 (7.7)
Total	46/292 (15.7)	15/173 (8.7)

**Table 5 ijerph-16-02113-t005:** Rodent species investigated for *Leptospira* species in South-East Asia, with an x indicating the presence of the *Leptospira* species in that specific rodent species.

Rodent Species	Infection % (#)	*Leptospira* Species					Test Method				Country
		*Borgpetersenii*	*Interrogans*	*Kirschneri*	*Weilii*	*Inadai* *	Serology	DNA (Kidney)	Culture	DNA (Isolates)	
*Bandicota bengalensis*	12% (*n* = 42)					x	x		x		India [99]
*Bandicota bengalensis*	17% (*n* = 42)	x *	x *				x		x		India [99]
*Bandicota bengalensis*	18.6% (*n* = 140)	x	x					x			Bangladesh ^†^
*Bandicota bengalensis*	16.6% (*n* = 58)		x				x				India [71]
*Bandicota indica*	4% (*n* = 75)	x *	x *			x	x		x		India [99]
*Bandicota indica*	23% (*n* = 75)						x		x		India [99]
*Bandicota indica*	3.7% (*n* = 27)		x					x			Thailand, Lao PDR, Cambodia [78]
*Bandicota indica*	2.7% (*n* = 36)		x						x	x	Thailand [100]
*Bandicota indica*	3.5% (*n* = 170)	x	x				x				Thailand [60,96]
*Bandicota indica*	10.8% (*n* = 65)	x	x					x			Vietnam [101]
*Bandicota indica*	77.8% (*n* = 9)	x	x					x			Bangladesh ^†^
*Bandicota savilei*	1.9% (*n* = 52)				x			x			Thailand, Lao PDR, Cambodia [78]
*Bandicota savilei*	0% (*n* = 2)								x	x	Thailand [100]
*Bandictoa savilei*	2.3% (*n* = 175)	x	x				x				Thailand [60,96]
*Rattus exulans*	0.45% (*n* = 220)		x					x			Thailand, Lao PDR, Cambodia [78]
*Rattus exulans*	18.2% (*n* = 11)		x					x	x		New Caledonia [92]
*Rattus exulans*	34.8% (*n* = 23)	x	x	x				x			Bangladesh ^†^
*Rattus exulans*	38% (*n* = 19)						x			x	Malaysia [68]
*Rattus exulans*	0% (*n* = 1)							x	x		Malaysia [102]
*Rattus exulans*	6.8% (*n* = 322)	x	x				x				Thailand [96]
*Rattus exulans*	6.9% (*n* = 317)	x	x				x	x	x		Thailand [60]
*Rattus exulans*	0% (*n* = 3)								x	x	Thailand [100]
*Rattus exulans*	14.3%		x				x				Vietnam [101]
*Rattus rattus*	17.8% (*n* = 129)		x					x	x		New Caledonia [92]
*Rattus rattus*	7.9% (*n* = 191)	x	x					x			Bangladesh ^†^
*Rattus rattus*	14.3% (*n* = 28)		x				x				India [71]
*Rattus rattus*	7.1% (*n* = 85)	x	x	x			x				Andaman Islands [70]
*Rattus rattus*	7% (*n* = 285)	x	x				x			x	Malaysia [68]
*Rattus rattus*	70% (*n* = 20)	x	x						x	x	Malaysia [79]
*Rattus rattus*	11.9% (*n* = 59)	x	x	x				x	x		Malaysia [102]
*Rattus rattus*	5% (*n* = 464)	x	x				x	x	x		Thailand [60]
*Rattus rattus*	4.7% (*n* = 492)	x	x				x				Thailand [96]
*Mus musculus*	5.5% (*n* = 97)	x	x					x			Bangladesh ^†^
*Mus musculus*	0% (*n* = 4)										Thailand [96]
*Mus terricolor*	0% (*n* = 5)							x			Bangladesh ^†^

* All isolates were tested with Microscopic Agglutination Test (MAT) assay on a reference panel of 21 well-known serovars, from the *L. interrogans* and *L. borgpetersenii* group, but also on 1 serovar from *L. noguchii, L. biflexa*, and *L. satarosai* groups—unfortunately, no specification on which serogroups tested positive is mentioned. ^†^ This study.

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
