# Peer review of "Prevalence of Leptospira Infection in Rodents from Bangladesh"

_ijerph, 2019, doi:10.3390/ijerph16122113_

Round 1

Reviewer 1 Report

Krijger et al. present an interesting study on the prevalence of Leptospira infections in rodents of Bangladesh. The correct presentation and interpretation of this study can provide the basis for extensive efforts in ameliorating the increasing risk of human leptospirosis. The study collected a valuable amount of data from different geographical locations in Bangladesh and compares the obtained results with other available results in Asia. In short, this study merits its continuation and after an extensive revision following the observations below, I think it should be published.

Nevertheless, I have to be honest and manifest my disappointment when I found in the text the description of the sections from the Author's Guidelines of the journal: end of the Materials and Methods section, which merges the description of the Statistical analysis and the description of what is Materials and Methods; the conclusions section is just the sentence of what should be written there; and Funding is just the statement of what should be written there. My intention is to remind the authors that you should read the final PDF or at least the final manuscript, if I am willingly reading it, I expect the authors to have read their own creation. For instance, there is a mention of Table 5 in line 358/359, but there is not Table 5 in the text.

The text can be improved by removing redundant ideas present in both the introduction and discussion, as both are lengthy in relation to the length of the results. The text should be revised to make it more concise, I suggest considering proof-reading services.

I would like to repeat that I consider this study has potential if it is properly amended. Following some constructive comments to consider on your revision:

-In materials and methods, study area: it would be good to include what months of 2016 and 2017 the sampling was performed, as it is described for March 2015 to March 2016, but not mentioned for the other years. It seems suspicious.

-In line with the previous comment, but as a possible addition: consider making available the distribution of collected rodents by region and month/year. As the discussion presents some points about the number of rodents in dry and rainy season and the relation with the number of infected rodents. Although Table 1 is informative as it is, a finer detail on the number of infected samples could help to extend more the findings of this study. Such data could be considered as supplementary material.

-In the materials and methods, it is described that captured rodents were dissected, if so, information of such dissection could enrich the results section. For example, the number of infected rodents presenting damage due to the corresponding Leptospira infection. If not damage was present, then the role as natural reservoir is confirmed.

-Include a reference to what section of the Leptospira was sequenced or the primers used for the sequencing and posterior identification of the serovars.

Author Response

We want to thank the reviewers for their constructive comments on this manuscript, it helped us to improve the manuscript. We took all the comments (listed below) into account and respond below to all the points raised by the reviewers (our reaction in italics). In addition, we incorporated all the proposed changes into the manuscript, by highlighting the text in the accompanying document using the ‘track changes’ function of Microsoft Word.

Rev 1

Krijger et al. present an interesting study on the prevalence of Leptospira infections in rodents of Bangladesh. The correct presentation and interpretation of this study can provide the basis for extensive efforts in ameliorating the increasing risk of human leptospirosis. The study collected a valuable amount of data from different geographical locations in Bangladesh and compares the obtained results with other available results in Asia. In short, this study merits its continuation and after an extensive revision following the observations below, I think it should be published.

Thank you. We did our very best to revise the manuscript according to your insights and suggestions.

Nevertheless, I have to be honest and manifest my disappointment when I found in the text the description of the sections from the Author's Guidelines of the journal: end of the Materials and Methods section, which merges the description of the Statistical analysis and the description of what is Materials and Methods; the conclusions section is just the sentence of what should be written there; and Funding is just the statement of what should be written there.

You are right. Our sincere apologies. This happened when we copied our text into the journals template. We agree it is sloppy and have re-read the final revised version.

My intention is to remind the authors that you should read the final PDF or at least the final manuscript, if I am willingly reading it, I expect the authors to have read their own creation. For instance, there is a mention of Table 5 in line 358/359, but there is not Table 5 in the text.

This is correct, we adjusted this mistake. It should have read Table 4, however, due to the addition of another Table (as you suggest later on) it now becomes Table 5.

The text can be improved by removing redundant ideas present in both the introduction and discussion, as both are lengthy in relation to the length of the results. The text should be revised to make it more concise, I suggest considering proof-reading services.

We edited the introduction and discussion section according this suggestion. We understand that some of the sections may seem to go into a lot of detail, but we consider this detail to be important if we are to really consider all aspects.

I would like to repeat that I consider this study has potential if it is properly amended. Following some constructive comments to consider on your revision:

In materials and methods, study area: it would be good to include what months of 2016 and 2017 the sampling was performed, as it is described for March 2015 to March 2016, but not mentioned for the other years. It seems suspicious.
Thank you for this comment, we added the months according your suggestion, please see line 127-130.

In line with the previous comment, but as a possible addition: consider making available the distribution of collected rodents by region and month/year. As the discussion presents some points about the number of rodents in dry and rainy season and the relation with the number of infected rodents. Although Table 1 is informative as it is, a finer detail on the number of infected samples could help to extend more the findings of this study. Such data could be considered as supplementary material.
We agree with this suggestion and added a table (Table 4) with information on infection per rodent species divided up to the dry and wet seasons. As supplemental data we added a Table (S1) with information on all positive samples (such as trapping date and location), mentioned in line 200.

In the materials and methods, it is described that captured rodents were dissected, if so, information of such dissection could enrich the results section. For example, the number of infected rodents presenting damage due to the corresponding Leptospira infection. If not damage was present, then the role as natural reservoir is confirmed.
We found no damage or anomalies during the dissections. So indeed, this confirms the role of the animals as natural reservoir. We added this in the manuscripts, see line 197 and 256

Include a reference to what section of the Leptospira was sequenced or the primers used for the sequencing and posterior identification of the serovars.
Serovar identification with molecular techniques is not (yet) possible. Up to now serovar identification is done by Cross Agglutinin Absorption Test, a serological method. We have done species identification in this manuscript. The PCR product was sequenced. The target gene is secY, the reference is already in the manuscript: Ahmed A, Engelberts MFM, Boer KR, Ahmed N, Hartskeerl RA (2009) Development and Validation of a Real-Time PCR for Detection of Pathogenic Leptospira Species in Clinical Materials PLoS ONE 4:e7093 doi:10.1371/journal.pone.0007093

Reviewer 2 Report

This paper presents interesting results about leptospira infections in rodents in Bangladesh. The abstract is satisfactory but the introduction is a bit repetitive and requires editing for grammatical errors. There is a section in the materials and methods which seems to be cut and pasted from the manuscript preparation guidelines (lines 171-181). The discussion contains some relevant insights but seems repetitive and requires editing for a wide range of grammatical errors. There is reference to Table 5 but I could not find this in the manuscript. The conclusions section is missing,  there seems to be a section cut and pasted from the manuscript guidelines (lines 389-390). 

Author Response

We want to thank the reviewers for their constructive comments on this manuscript, it helped us to improve the manuscript. We took all the comments (listed below) into account and respond below to all the points raised by the reviewers (our reaction in italics). In addition, we incorporated all the proposed changes into the manuscript, by highlighting the text in the accompanying document using the ‘track changes’ function of Microsoft Word.

Rev 2

This paper presents interesting results about leptospira infections in rodents in Bangladesh.

Thank you.

The abstract is satisfactory but the introduction is a bit repetitive and requires editing for grammatical errors. There is a section in the materials and methods which seems to be cut and pasted from the manuscript preparation guidelines (lines 171-181).

Our sincere apologies. Both you and the other reviewer pointed out this sloppy mistake and we immediately adjusted this. The mistake happened when we copied our text into the journals template. We have re-read the final revised version to be sure no mistakes of this kind were made anymore.

The discussion contains some relevant insights but seems repetitive and requires editing for a wide range of grammatical errors.

We edited the discussion section according this suggestion. We understand that some of the sections may seem lengthy, but we consider this to be important if we are to really consider all aspects.

There is reference to Table 5 but I could not find this in the manuscript.

This is correct, we adjusted this mistake. It should have read Table 4, however, due to the addition of another Table -as per suggestion of Reviewer 1- it now becomes Table 5.

The conclusions section is missing, there seems to be a section cut and pasted from the manuscript guidelines (lines 389-390).

Our sincere apologies, we adjusted this, please see line 426-443

Round 2

Reviewer 1 Report

I am glad to read the text after it was properly amended, and that my suggestions helped to improve the manuscript. My final suggestion is to make sure the references properly match the text after removing some sections of the text, for example lines 82-87.